# Development and Clinical Performance of InteliSwab^®^ COVID-19 Rapid Test: Evaluation of Antigen Test for the Diagnosis of SARS-CoV-2 and Analytical Sensitivity to Detect Variants of Concern Including Omicron and Subvariants

**DOI:** 10.3390/v16010061

**Published:** 2023-12-29

**Authors:** Mark J. Fischl, Janean Young, Keith Kardos, Michele Roehler, Tiffany Miller, Melinda Wooten, Natalie Holmes, Nicole Gula, Mia Baglivo, Justin Steen, Nori Zelenz, Antony George Joyee, Vincent Munster, Zack Weishampel, Claude Kwe Yinda, Kevin G. Rouse, Cathy Gvozden, David Wever, Giralt Yanez, Marc Anderson, Song Yu, Brian Bearie, Stephen Young, Jody D. Berry

**Affiliations:** 1OraSure Technologies, Inc., 220 East First St., Bethlehem, PA 18015, USA; jyoung@orasure.com (J.Y.); mroehler@orasure.com (M.R.); tmiller@orasure.com (T.M.); mwooten@orasure.com (M.W.); mbaglivo@orasure.com (M.B.); joyeeag@yahoo.com (A.G.J.); immunoreagent@gmail.com (J.D.B.); 2Laboratory of Virology, Division of Intramural Research, National Institute of Allergy and Infectious Diseases, Hamilton, MT 59840, USA; vincent.munster@nih.gov (V.M.); zack.weishampel@nih.gov (Z.W.); yinda.kweclaude@nih.gov (C.K.Y.); 3The Children’s Clinic, Jonesboro, AR 72401, USA; krouse@jbrkids.com; 4Gvozden Pediatrics, Millersville, MD 21108, USA; 5Cahaba Research Inc., Pelham, AL 35124, USA; dwever@cahabaresearch.com; 6South Florida Research Organization, Medley, FL 33166, USA; yanez.sfro@gmail.com; 7Tanner Clinic, Layton, UT 84041, USA; 8Cahaba Research Inc., MedHelp Urgent Care, Birmingham, AL 32535, USA; syu@cahabaresearch.com; 9Urgent Care of Colton, Benchmark Research Group, Colton, CA 92324, USA; 10TriCore Reference Laboratories, Albuquerque, NM 87102, USA; steve.young@tricore.org

**Keywords:** COVID-19, SARS-CoV-2, rapid antigen test, diagnosis, Omicron, variants of concern

## Abstract

Background and objectives: Timely detection of SARS-CoV-2 infection with subsequent contact tracing and rapid isolation are considered critical to containing the pandemic, which continues with the emergence of new variants. Hence, there is an ongoing need for reliable point-of-care antigen rapid diagnostic tests (Ag-RDT). This report describes the development, evaluation, and analytical sensitivity of the diagnostic performance of the InteliSwab^®^ COVID-19 Rapid Test. Methods**:** Samples from 165 symptomatic subjects were tested with InteliSwab^®^ and the results were compared to RT-PCR to determine the antigen test performance. The analytical sensitivity of InteliSwab^®^ for the detection of different variants was assessed by limit of detection (LOD) determination using recombinant nucleocapsid proteins (NPs) and testing with virus isolates. Western immunoblot independently confirmed that each monoclonal Ab is capable of binding to all variants tested thus far. Results: The overall positivity rate by RT-PCR was 37% for the 165 symptomatic subjects. Based on RT-PCR results as the reference standard, InteliSwab^®^ showed clinical sensitivity and specificity of 85.2% (95% CI, 74.3–92.0%) and 98.1% (95% CI, 93.3–99.7%), respectively. The overall agreement was 93.3% (Kappa index value 0.85; 95% CI, 0.77–0.74) between RT-PCR and InteliSwab^®^ test results. Furthermore, the evaluation of analytical sensitivity for different SARS-CoV-2 variants by InteliSwab^®^ was comparable in the detection of all the variants tested, including Omicron subvariants, BA.4, BA.5, and BQ.1. Conclusions: Due to the surge of infections caused by different variants from time to time, there is a critical need to evaluate the sensitivity of rapid antigen-detecting tests for new variants. The study findings showed the robust diagnostic performance of InteliSwab^®^ and analytical sensitivity in detecting different SARS-CoV-2 variants, including the Omicron subvariants. With the integrated swab and excellent sensitivity and variant detection, this test has high potential as a point-of-care Ag-RDT in various settings when molecular assays are in limited supply and rapid diagnosis of SARS-CoV-2 is necessary.

## 1. Introduction

The COVID-19 pandemic, due to severe acute respiratory syndrome coronavirus 2 (SARS-CoV-2), caused millions of deaths globally, and infections continue to spread across the world, fueled by novel variants. Timely detection of SARS-CoV-2 infection with subsequent contact tracing and rapid isolation are considered critical to containing the pandemic [1,2]. Hence, there is an ongoing consistent need for rapid, accurate, and easy-to-perform diagnostic tests that can be used to test large numbers of individuals in a short period of time and in isolation in their own homes.

Currently, nucleic acid amplification tests (NAATs), such as RT-PCR assays, remain the gold standard for the diagnosis of SARS-CoV-2 infection [3,4]. Although molecular methods are highly sensitive and specific, they are expensive, time consuming, and require dedicated molecular laboratory setup and trained laboratory personnel. The need to test a large number of patients/contacts in a timely manner and frequently is often not satisfied due to limited laboratory capacities and slow turn-around time for obtaining RT-PCR results, which in turn can hinder public health containment strategies [5]. Therefore, there is a critical demand for faster, inexpensive, and easier-to-use alternative diagnostic tools such as novel point-of-care antigen-detecting rapid diagnostic tests (Ag-RDT) [6,7]. Early in the SARS-CoV-2 pandemic, neither the biology nor the immune response to the virus was well understood. Past experience with SARS-CoV strongly pointed to the spike protein as the target of neutralizing antibodies [8,9]. Similarly, the nucleocapsid protein (NP), less variable than the spike due to its relative abundance [10], was favored as the target for antigen detection in the face of antigenic variation. Although in general, Ag-RDTs are less sensitive than the NAATs, they may be particularly useful during the first several days of infection following the onset of symptoms and for detecting infectious individuals with greater viral load, thereby reducing the chances of transmission of the infection [11]. OraSure Technologies, Inc., in collaboration with the Biomedical Advanced Research and Development Authority (BARDA), began development on InteliSwab^®^ with the goal of creating a test that could detect the most infectious individuals. It is important to note that frequent testing is critical to controlling the spread of SARS-CoV-2 infections considering the infection kinetics and viral load changes [12]. This could be effective with point-of-care antigen tests, which are easy to perform and inexpensive, as repeated testing is practical and test results are available immediately.

Multiple rapid Ag-RDTs are now commercially available or in development. The InteliSwab^®^ COVID-19 Rapid Test (OraSure Technologies, Inc., Bethlehem, PA, USA) is a novel rapid Ag-RDT that uses a sandwich capture lateral flow immunoassay to detect SARS-CoV-2 nucleocapsid antigen in anterior nasal samples. It has a unique design that incorporates a fully integrated swab to directly collect samples from the anterior nasal cavity, thereby significantly simplifying the entire testing process. InteliSwab^®^ received Emergency Use Authorizations (EUAs) from the U.S. Food and Drug Administration (FDA) for three versions: professional use in point-of-care (POC) settings, over the counter (OTC) use, and prescription (Rx) home use. Here, OraSure reports the evaluation of InteliSwab^®^ performance for the detection of SARS-CoV-2 infection in symptomatic subjects in comparison with RT-PCR results as the reference standard. InteliSwab^®^ is also authorized for individuals without symptoms when tested at least three times over five days with at least 48 h between tests. In addition, the analytical sensitivity of InteliSwab^®^ for the detection of different SARS-CoV-2 variants, including Omicron and its subvariants, has been assessed.

## 2. Methods

### 2.1. Study Subjects, Sample Collection, and SARS-CoV-2 Testing

A prospective clinical study was conducted to evaluate the performance of the InteliSwab^®^ COVID-19 Rapid Test in six geographically diverse sites across the U.S. (the sites were in Jonesboro, AK; Pelham, AL; Medley, FL; Birmingham, AL; Colton, CA; and Millersville, MD). Individuals who met the inclusion criteria were consecutively included in the study. Inclusion criteria include the following, along with written informed consent: Subjects must be ≥2 years of age; subjects are exhibiting one or more of the following signs and symptoms on the day of study: fever, cough, shortness of breath, difficulty breathing, muscle pain, headache, sore throat, chills, repeated shaking with chills, new loss of taste or smell, congestion or runny nose, diarrhea, nausea or vomiting; and subjects report symptom onset within the past seven (7) days.

The sample size for the antigen test performance evaluation was determined based on the FDA EUA guidance documents. In this report, the evaluation included samples from two rounds of testing. The first round included 146 individuals (>15 years of age) with signs and symptoms of COVID-19 within the first seven days of symptom onset who completed the study (February through April 2021) and obtained a valid result. The subjects aged 18 years and older independently collected an anterior nasal sample, conducted the test, and reported their self-test result. For subjects aged 15–17 years, the parent/legal guardian collected the nasal sample, conducted the test, and interpreted and recorded the test result. An additional round of testing was conducted (August through September 2021) that consisted of 19 children aged 2–14 years, where the parent/legal guardian collected the nasal sample and performed the test. The study and the informed consent form were approved by the WCG Institutional Review Board. If the antigen test results were incomplete or not performed, or if lab results were missing for any samples, they were excluded from the analysis. A total of 165 samples from the two rounds of testing were combined to evaluate the diagnostic performance of the InteliSwab^®^ COVID-19 Rapid Test.

SARS-CoV-2 infection in this symptomatic study population was determined using a combination of highly sensitive molecular FDA EUA SARS-CoV-2 assays (Quidel Lyra SARS-CoV-2 Assay, Roche cobas^®^ SARS-CoV-2, and CDC 2019 nCoV Real-Time RT-PCR Diagnostic Panel). The Roche cobas^®^ assay (Roche Diagnostics, Indianapolis, IN USA) and the Quidel Lyra assay (Quidel, San Diego, CA, USA) were run on all samples. Any samples with discordant results between the Quidel Lyra and Roche cobas^®^ assays were run with the CDC assay per the package insert to obtain the final composite PCR result, which was taken as the reference standard to evaluate the antigen test performance. At the time of performing the antigen test, the performers/readers did not have reference method test results, as the reference method specimens were sent to Tricore Reference Laboratories, New Mexicofor testing. There were no adverse events in the performance study.

### 2.2. Design, Expression, and Purification of Recombinant NP Proteins from Variants of Concern (VOCs) and InteliSwab^®^ Testing

Nucleocapsid proteins of different SARS-CoV-2 variants were expressed with an N-terminus poly-histidine tag in BL21(DE3) cells and purified by affinity capture of the proteins using nickel sepharose HP as described [13]. All recombinant NPs used for device testing had >95% purity. All of the testing procedures were carried out as described previously [13]. Briefly, to load the sample onto the test device, 50 µL of sample were added to the center of the flat pad. With the developer solution vials positioned in the test stand, the loaded test devices were placed in separate vials containing the developer solution. Each test device remained in the developer solution for 30 min, after which the result was read. A positive result was indicated by a reddish-purple line at the test and control zones.

### 2.3. Western Immunoblot of Recombinant NP Proteins Showing Individual Monoclonal Antibodies Recognize All the NPs of VOC to Date

Western immunoblot is a qualitative and orthogonal way to confirm antibody reactivity and is distinct from lateral flow. Equivalent amounts of NPs from the VOCs were loaded in sample buffer with SDS (no reducing agents) in each lane of a 4–12% Bis Tris gel, as shown in Figure 1. NP is comprised of 419 amino acids and via SDS-PAGE is approximately 55 kDa. Proteins were transferred to a nitrocellulose membrane for 7 min using the iBlot transfer system. Membranes were blocked with dehydrated milk/TBST. Monoclonal antibody 1 was applied at 50 ng/mL in milk/TBST and 30 ng/mL in milk/TBST for antibody 2. Primary incubation was conducted at room temperature for 30 min and washed 3X with TBST. Secondary detection was conducted via a standard secondary antibody labeled with alkaline phosphatase and developed with NBT/BCIP. Images were captured using an iPhone 11.

### 2.4. InteliSwab^®^ COVID-19 Rapid Test on SARS-CoV-2 Isolates

Tests with isolates of different SARS-CoV-2 variants were performed as previously described [13]. Briefly, 10-fold serial dilutions of irradiated SARS-CoV-2 variants were performed and 3 InteliSwab^®^ tests were used to test each dilution, making a total of 15 test devices per variant. More precise limits of detection (LODs) for the test were determined by performing 2-fold serial dilutions in three replicates for each variant, starting with the highest concentration used for the initial 10-fold dilution series. The LOD was confirmed as the lowest concentration of recombinant protein that was detected ≥ 95% of the time (i.e., at a concentration where 19 out of 20 test results were positive).

### 2.5. Analytical Sensitivity and Limit of Detection (LOD) Determination

The analytical performance of InteliSwab^®^ was assessed by testing the LOD with different concentrations of recombinant SARS-CoV-2 nucleocapsid protein diluted in synthetic nasal matrix. Due to the surge of infections caused by different variants from time to time, there is a critical need to evaluate the sensitivity of rapid antigen-detecting tests for new variants. Therefore, the LODs of InteliSwab^®^ were analyzed for the different variants, Alpha, Beta, Gamma, Delta, Lambda, Mu, and Omicron and its subvariants, BA.2, BA.4, BA.5, and BQ.1, along with the parental strain, using corresponding recombinant NPs and contrived samples spiked with whole inactivated virus isolates. The contrived samples were randomized, and operators were blinded to the sample identities for testing on the InteliSwab^®^ test. The LOD was confirmed as the lowest concentration of recombinant protein that was detected ≥ 95% of the time (i.e., at a concentration where 19 out of 20 test results were positive). All InteliSwab^®^ tests were read at 30 min.

### 2.6. Statistical Analysis

Statistical analysis was performed using GraphPad Prism 9 (GraphPad Software, Inc., Boston, MA, USA) and Microsoft Excel 360 (Microsoft Corp., Redmond, WA, USA). The performance of the InteliSwab^®^ COVID-19 Rapid Test was evaluated by taking RT-PCR positivity as the reference standard, and two-by-two cross-tab analysis was used for the calculation of sensitivity (positive percent agreement (PPA)), specificity (negative percent agreement (NPA)), positive predictive value (PPV), and negative predictive value (NPV). The positive and negative predictive values were determined while taking RT-PCR positivity into account. To determine the concordance of diagnostic performance, the Cohen’s kappa (κ) value was used and interpreted according to the criteria proposed by Landis and Koch [14].

### 2.7. Reporting Guidelines

STARD reporting guidelines were used for completeness and transparency of reporting diagnostic accuracy studies [15].

## 3. Results

### 3.1. Clinical Scouting with Earlier Prototypes

Test and collection procedure indications were varied during prototype development, starting with testing oral fluid and moving towards a nasal swab collection. Early prototypes utilized a polyclonal anti-SARS-CoV-2 NP antibody. As reagents were optimized in real time, successive prototypes comprised new monoclonal antibodies against SARS-CoV-2 NP.

The device has an integrated nasal swab and utilizes conformational high-affinity monoclonal antibodies that confer greater resistance to individual point mutations (antigenic variation) in variants and have far superior sensitivity (>150-fold) to that of the polyclonal antibody in lateral flow. Rapid iteration of prototypes to the current final version of InteliSwab^®^ was possible due to the team’s prior experience in creating the OraQuick^®^ Ebola antigen test, OraQuick^®^ HCV test, and OraQuick^®^ Advance HIV-1/2 test.

### 3.2. Western Immunoblot of NPs from Variants as Well as VOCs with Individual Monoclonal Antibodies (mAbs) Used in InteliSwab^®^

Western blot under non-reducing conditions clearly shows that each monoclonal antibody used in InteliSwab^®^ can detect all VOCs as well as the original Wuhan NP, 2020–2021 (Figure 1). This qualitative orthogonal approach to assessing sensitivity and specificity correlates with the assembled device data. Variants are defined and grouped based on the mutations in the spike protein. SARS-CoV-2 NPs are more conserved than spike proteins in regards to mutational evolution; therefore, NP is an optimal target for the assay. Western blot data in Figure 1 show that antibodies used in the test are qualitatively equivalent in variant detection across the 10 variants of concern tested. We can conclude that epitopes for each antibody are not affected by the amino acid substitutions. The lack of adverse affects of antibody binding to variants of SARS-CoV-2 may be in part due to the antibodies recognizing conformational epitopes. Amino acids involved in mutating may be buried residues or located in disordered regions of the protein and therefore and do not affect the overall folding of the protein. MERS-CoV and SARS-CoV-2 NP were tested for cross-reactivity. The results show that both antibodies had specificity only to SARS-CoV-2 NP and variants of SARS-CoV-2 and did not react with MERS or SARS-CoV NP (Figure 1).

### 3.3. Clinical Performance Evaluation of InteliSwab^®^

Samples from a total of 165 symptomatic individuals were tested on InteliSwab^®^. The overall positivity for COVID-19 in this study was determined to be 37% (61/165) based on RT-PCR results. The InteliSwab^®^ showed positivity in 52 out of the 61 RT-PCR positive samples. Eight out of nine samples that were InteliSwab^®^ false negative had RT-PCR Ct values greater than 30, which correlate with low viral titer. Low viral titer will result in lower prevalence of SARS-CoV-2 antigens overall, thereby providing an explanation for the difference in positive results between the two tests. Among the 104 RT-PCR negative samples, 102 samples were negative via InteliSwab^®^. Potential reasons for the two false negative InteliSwab^®^ results could be due to sample collection or variability among types of sample collection between InteliSwab^®^ and RT-PCR. In order to comment further on the differences between RT-PCR results vs. InteliSwab^®^ results, a larger controlled study would be required. Using RT-PCR results as the reference standard, InteliSwab^®^ yielded sensitivity and specificity of 85% and 98%, respectively (Table 1). The overall agreement between the RT-PCR and InteliSwab^®^ was 93.3%, with a Cohen’s Kappa (ⱪ) value of 0.85 (95% CI, 0.77–0.74), indicating high correlation between the two test results. The antigen test positivity rate observed with different age groups is shown in Table 2.

Further, the correlation between InteliSwab^®^ test positivity and PCR cycle threshold (Ct) values was analyzed. Ct values were stratified into four groups—<20, 21– 25, 26–30, and 31–35—and accordingly, the antigen test results were compared to RT-PCR positivity (Table 3). In positive subjects whose Cobas^®^ PCR result had a Ct value of up to 25, InteliSwab^®^ showed high sensitivity at 97.7% (42/43), though one sample with a low Ct value of 19 was not detected. Although it is still unclear, in general, false negativity from antigen tests could also occur due to factors including endogenous inhibitors, antigen degradation, etc. (Note that, for all subjects, use of the test was monitored to ensure proper collection of the nasal swabs, and all test procedures were performed correctly). At higher Ct values, as expected, the sensitivity of InteliSwab^®^ decreased (Table 3). PCR positive samples were in the higher Ct value range (27–35) for eight out of nine antigen tests, suggesting that the negative results could have been due to low viral loads.

### 3.4. Evaluation of Analytical Sensitivity Using Recombinant NP and Virus Isolates

To assess whether the detection ability of InteliSwab^®^ is affected by different SARS-CoV-2 variants, the detection limits for each variant were tested using quantified highly pure recombinant NP and virus isolates [13] as well as the Omicron subvariants. The LODs were determined for different variants—Alpha, Beta, Gamma, Delta, Lambda, Mu, and Omicron BA.1, BA.2, BA.4, BA.5, and BQ.1—along with that of the parental Wuhan strain using corresponding NPs. The LOD for the parental strain, Beta, Delta, Lambda, Mu, and Omicron (BA.1) variants (Table 4) was determined to be 0.469 ng/mL. For Alpha strains, Gamma, and the Omicron subvariants BA.2, BA.4, BA.5, and BQ.1, the LOD was found to be 0.313 ng/mL. Furthermore, testing using cell-cultured SARS-CoV-2 variants led to the observation of only minor differences in the sensitivity of InteliSwab^®^ for Alpha, Beta, Gamma, Delta, and Omicron and its subvariants BA.2, BA.4, and BA.5 compared to the initial lineage A variant (Table 4). Overall, the testing results showed comparable analytical sensitivity for all of the different variants evaluated.

## 4. Discussion

In the present study, we evaluated the diagnostic accuracy of InteliSwab^®^, a rapid SARS-CoV-2 antigen test for the detection of SARS-CoV-2, in a group of symptomatic subjects. In addition, the analytical performance of InteliSwab^®^ for the detection of variants was assessed. The results show excellent clinical performance of InteliSwab^®^ (85.2% sensitivity and 98.1% specificity) compared with the reference method, RT-PCR, and meets the WHO recommendation of ≥80% sensitivity and ≥97% specificity [16], as with other commercially available Ag-RDTs [6,17,18,19]. However, this only applies in symptomatic patients with suspected COVID-19 who are tested shortly after symptom onset (up to 7 days in the current study). Therefore, it is important to further evaluate the performance of InteliSwab^®^ in asymptomatic and other community settings, and such studies are underway. The Ct values of RT-PCR could be considered as a semi-quantitative correlate for viral load and could be correlated to the ability of detection of the Ag-RDT. As expected, in samples with lower Ct values, InteliSwab^®^ showed higher sensitivity. InteliSwab^®^ detected most of the samples that were positive via PCR at a Ct value ≤ 25, whereas the sensitivity decreased in samples with a high Ct value. This shows that InteliSwab^®^ can identify individuals with high viral load. Therefore, the use of antigen tests can be very helpful to identify and quarantine contagious individuals and to trace contacts. In this regard, the ease of use and the short time to obtain the test results can be very advantageous for Ag-RDTs like InteliSwab^®^, especially in resource-limited/field settings. The design of InteliSwab^®^ with an integrated swab provides simplicity and ease of testing. Usability studies demonstrated that the error rate during the performance of the assay was very minimal. The COVID-19 pandemic is ongoing, and infections continue to spread around the world with the evolution of the virus and surges in infections due to new SARS-CoV-2 variants. Highly contagious Omicron subvariants such as BA.5, BQ.1/BQ 1.1, and XBB have been causing new waves of infection around the world. Further, periodic surges of COVID-19 in China and the continuous spread of infections underscore the importance of rapid diagnosis and containment. Therefore, reliable Ag-RDTs are invaluable for the timely detection of SARS-CoV-2 infection with subsequent contact tracing and rapid isolation.

Mutations of the spike protein are used to define a variant. Indeed, NP most often has few or little changes among variants and quite often is identical between variants of concern. However, with the continuous evolution of virus variants, there is a concern that new variants can evade COVID-19 tests and provide inaccurate results. Therefore, it is important that current tests, including Ag-RDTs, be evaluated constantly to ensure their sensitivity in detecting new variants. Previously, analytical studies with recombinant NP and whole-virus isolates up until the Omicron BA.1 variant [13] were performed. In the present study, testing was expanded using these methods to test the ability of InteliSwab^®^ to detect different SARS-CoV-2 variants, including the new Omicron subvariants. The overall findings suggest that the ability of InteliSwab^®^ to detect variants was not affected. InteliSwab^®^ showed comparable sensitivity of detection for the different variants, including Omicron and its subvariants. The ultimate performance of the rapid antigen test largely depends on the antibodies used in the device to bind to the NP antigen. In this respect, the binding ability of the mAbs used in InteliSwab^®^ against NPs of different variants was validated. This was confirmed and visualized independently using Western blot under non-reducing conditions that clearly show that each monoclonal antibody can bind well to all VOCs as well as to Wuhan-Hu-1 2019 Accession NC_045512 NP. (Figure 1) Of note, the two mAbs used for detection in the device showed characteristics of being conformational mAbs and bound parts of NPs that were spatially distant in the linear sequence. Analysis of these two mAbs in Pepscan using 15 mers failed to identify any linear stretches important enough to allow binding (PEPperPRINT, negative). It is interesting that they also work well in Western immunoblot, which was long thought to be a method of verification of more “linear-like” epitope binding, at least under reducing conditions [20]. The SARS-CoV-2 NP does not have cysteine residues naturally (except in occasional variants); therefore, reducing agents in the gel and sample were excluded. Lane 8 of both immunoblots (Figure 1) shows the recombinant NP of a variant that had a cysteine residue arise via mutation, which explains the clear dimer under non-reducing conditions. This conformational property likely makes them strong candidates to withstand individual point mutations in the NP, as single mutations are unlikely to affect all contact residues from continuing to be bound by the mAb. Taken together, the results show the broad reactivity of the mAbs, which is in line with the high analytical sensitivity of the InteliSwab^®^ antigen test for the detection of all of the variants of SARS-CoV-2 tested thus far. To our knowledge, this is the first report on an Ag-RDT to evaluate the analytical sensitivity of detection of recent Omicron subvariants, including BA.4, BA.5, and BQ.1. The NP mutations of XBB/XBB 1.5 are the same as those of BA.2; therefore, it was not required to test those subvariants.

In addition to the strains named throughout the manuscript, OraSure has concluded that variants of concern/variants of interest will have data equivalent or similar to the following: Omicron, B.1.1.529, BA.1, BA.2.12.1, BQ.1.1, BA.3, BA.4.6, EG.5, EG.5.1, FL.1.5.1, and BA.2.86. InteliSwab^®^ performance was assessed by either wet testing or in silico analysis. Wet testing was conducted using recombinant proteins containing the relevant mutations as well as live viral isolates of the relevant strains. Wet testing was conducted at OraSure and by third-party laboratories, including a collaborative study with the NIH demonstrating that InteliSwab^®^ detected the Omicron variant in live clinical samples.

There are some limitations, as the study population includes only symptomatic individuals and the performance in intended use with asymptomatic individuals was not evaluated herein. It is known that most Ag-RDTs in general have shown reduced sensitivity in asymptomatic populations [12,16,19,21], but this can be mitigated by serial testing [22]. Although the laboratory-based findings indicate that InteliSwab^®^ can detect all different variants, including VOCs, it is important to have detailed clinical studies in real-world settings. Nevertheless, despite slight differences in detection limits, Ag-RDTs can serve as an important diagnostic tool for controlling the spread of SARS-CoV-2. Based on the current and previous results [13], recombinant NP analysis could be a valuable and practical analytical tool to enable routine monitoring of the function of Ag-RDTs and be used as a surrogate to BSL-3 live viral testing.

Overall, the present study’s findings showed that the InteliSwab^®^ COVID-19 Rapid Test exhibited excellent diagnostic performance and analytical sensitivity in detecting variants, including Omicron [13,23] and its subvariants, demonstrating its utility as a good point-of-care Ag-RDT in various settings when molecular assays are limited and rapid diagnosis of SARS-CoV-2 is necessary.

## Figures and Tables

**Figure 1 viruses-16-00061-f001:**
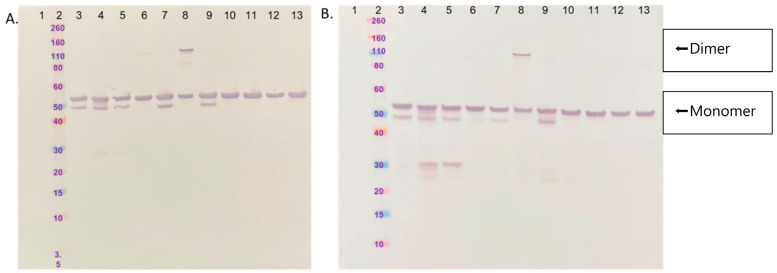
Detection of SARS-CoV-2 VOC nucleocapsid protein by Western blot. (**A**) Western Blot analysis of OTI SARS-CoV-2 NP variants with mAb 1 (**A**) and mAb 2 (**B**) from InteliSwab^®^. Recombinant MERS-CoV and SARS-CoV-2 NPs were analyzed for cross-reactivity. SARS-CoV-2 NP comprises 419 amino acids and migrates to approximately 55 kDa by SDS-PAGE. Lanes/samples are as follows: (1) MERS-CoV NP; (2) standards; (3) SARS-CoV-2 Wuhan NP; (4) SARS-CoV-2 B.1.17 NP (with 2 mutations in NP); (5) SARS-CoV-2 B.1.1.7 NP (with 4 mutations in NP); (6) SARS-CoV-2 B.1.617.2 NP; (7) SARS-CoV-2 P80R NP; (8) SARS-CoV-2 C.37 mutation (addition of a Cys G214C), hence, dimer is observed; (9) SARS-CoV-2 B.1.621 NP; (10) SARS-CoV-2 BA.1 NP; (11) SARS-CoV-2 BA.2 NP; (12) SARS-CoV-2 BA.4 NP; (13) SARS-CoV-2 BQ.1 NP.

**Table 1 viruses-16-00061-t001:** Performance characteristics of InteliSwab^®^ assay based on RT-PCR results *.

InteliSwab^®^ Assay	Reference RT-PCR *		
	Positive	Negative	Subtotal
Positive	52	2	54
Negative	9	102	111
Subtotal	61	104	165
Sensitivity/PPA	85.2% (95% CI, 74.3–92.0%)		
Specificity/NPA	98.1% (95% CI, 93.3–99.7%)		
Positive predictive value (PPV)	96.3% (95% CI, 86.8% to 99.0%)		
Negative predictive value (NPV)	91.9% (95% CI, 86.1–95.4%)		
Accuracy	93.3% (95% CI, 88.4–96.6%)		

* For reference standard, RT-PCR positive and negative results were determined by a composite algorithm, as described in the Methods section.

**Table 2 viruses-16-00061-t002:** Sample positivity by InteliSwab^®^ in different age groups.

	Positivity InteliSwab^®^	
Age Group (Years)	Number of Specimens	Positivity Number (Percentage)
2–13	19	9 (47.4%)
14–23	26	11 (42.3%)
24–64	111	33 (29.7%)
65+	9	1 (11.1%)
Total	165	54 (32.7%)

**Table 3 viruses-16-00061-t003:** Sensitivity of the InteliSwab^®^ Rapid Test based on RT-PCR Ct intervals.

Ct Values	InteliSwab^®^ Sensitivity (CI 95%)
≤20	97% (28/29) (95% CI: 83–99%)
21–25	100% (14/14) (95% CI: 78–100%)
26–30	73% (8/11) (95% CI: 43–90%)
31–35	17% (1/6) (95% CI: 3–56%)
Overall	85% (51/60) (95% CI: 74–92%)

**Table 4 viruses-16-00061-t004:** Analytical sensitivity of InteliSwab^®^ for detection with recombinant nucleocapsid protein (NP) and virus isolates.

SARS-CoV-2 Nucleocapsid Protein (NP) Variants	NP Mutation	Limit of Detection Recombinant NP	TCID_50_/mL (Virus Isolates)
Wuhan (WA1)	N/A	0.469 ng/mL	2.5 × 10^2^
Gamma	P80R	0.313 ng/mL	2.5 × 10^3^
Beta (B.1.351)	T205I	0.469 ng/mL	5 × 10^2^
Alpha (B.1.1.7)	D3L, S235F	0.313 ng/mL	2.5 × 10^2^
Alpha (B.1.1.7)	D3L, S235F, R203K, G204R	0.313 ng/mL	2.5 × 10^2^
Lambda (C.37)	G214C	0.469 ng/mL	NT
Mu (B.1.621)	T205I	0.469 ng/mL	NT
Delta (B.1.617.2)	D63G/R203M/D377Y	0.469 ng/mL	5 × 10^2^
Omicron BA.1	P13L, Deletions E31, E32, E33, R203K, G204R	0.469 ng/mL	5 × 10^2^
Omicron BA.2, XBB, XBB 1.5	P13L, deletions E31-33, R203K, G204R, S413R	0.313 ng/mL	5 × 10^2^
Omicron BA.4	P13L, deletions E31-33, R203K, G204R, S413R, P151S	0.313 ng/mL	1 × 10^2^
Omicron/BA.5	P13L, deletions E31-33, R203K, G204R, S413R	0.313 ng/mL	1 × 10^3^
Omicron/BQ.1 and Omicron/BQ.1.1	P13L, deletions E31-33, R203K, G204R, E136D, S413R	0.313 ng/mL	NT

NT, not tested.

## Data Availability

The study design is described in the text and additional details can be provided upon reasonable request.

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
