# Peer review of "Development and Clinical Performance of InteliSwab^®^ COVID-19 Rapid Test: Evaluation of Antigen Test for the Diagnosis of SARS-CoV-2 and Analytical Sensitivity to Detect Variants of Concern Including Omicron and Subvariants"

_viruses, 2023, doi:10.3390/v16010061_

Round 1

Reviewer 1 Report

Comments and Suggestions for Authors

This is a paper for analytical and clinical peformance of RAT for COVID-19 and Omicron variants. I attached the word file for the peer review. 

Please be careful to use a scientific terminology.

Comments on the Quality of English Language

Quality of English is excellent, but illustration of Fig and Tables are not good enough. 

Reviewer 2 Report

Comments and Suggestions for Authors

The article represents the study well however I recommend that Authors describe the methods in detail further. Moreover it would be better if the authors can discuss the results presented further. Few comments are below:

1. Authors should include more discussion regarding Figure 1. The stain behaviour observed in Western Blot analysis of variants with mAb 1 and mAb2 should explain in detail

2. Samples from a total of 165 symptomatic individuals were tested on InteliSwab®. The overall positivity for COVID-19 in this study was determined to be 37% (61/165) based on RT-PCR results. The InteliSwab® showed positivity in 52 out of the 61 RT-PCR positive samples. Autohrs should discuss the reason behind the positive and negative errors. Among the 104 RT-PCR negative samples, 102 samples were negative by InteliSwab®.

3. The LOD for the parental strain, Beta, Delta, Lambda, Mu, and Omicron (BA.1) variants (Table 4) was determined to be 0.469 ng/mL. For Alpha strains, Gamma, and the Omicron subvariants, BA.2, BA.4, BA.5 and BQ.1 the LOD was found to be 0.313 ng/mL. Authors should mention how they calculated the LOD values.
